# Associated factors with adherence to preventive behaviors related to COVID-19 among medical students in the university of Monastir, Tunisia

Imen Zemni [1,2,3]*, Kaouthar Zemni[1,4], Amal Gara[1,2], Amani Maatouk[1,2], Manel Ben Fredj[1,2,3], Hela Abroug[1,2,3], Meriem Kacem[1,2,3], Cyrine Benasrallah[1,2,3], Wafa Dhouib[1,2,3], Ines Bouanene[1,2], Asma Belguith Sriha[1,2,3]

1 Department of Epidemiology and Preventive Medicine, Fattouma Bourguiba University Hospital, University of Monastir, Monastir, Tunisia, 2 Department of Epidemiology, Faculty of Medicine of Monastir, University of Monastir, Monastir, Tunisia, 3 Technology and Medical Imaging Research Laboratory—LTIM—LR12ES06, University of Monastir, Monastir, Tunisia, 4 Department of Family Medicine, Faculty of Medicine of Monastir, University of Monastir, Monastir, Tunisia

* imen11zemni@gmail.com

**Data Availability Statement:** All relevant data are within the paper and its Supporting Information files.

## Abstract

### Introduction

Medical students should act as a model for the community in terms of compliance with preventive practices toward COVID-19. The aim of this study was to assess adherence to preventive behaviors related to COVID-19 among medical students and to identify its associated factors.

### Population and methods

We conducted a cross-sectional survey in October 2020 in the faculty of medicine of Monastir. We included a representative sample of medical students during registration days for the 2020–2021 academic year. The data were collected through a self-administered anonymous questionnaire. Eleven items related to preventive practices against COVID-19 were assessed (respiratory hygiene practices (Six Item), hand hygiene practices (Three Items) and social distancing (two items)). Items were evaluated using a Likert scale of five points (from 0: (Never) to 4: (Always)). The score obtained from the sum of these items allowed to classify students into two categories: "Good compliance" if the score was $\geq$ 80% and "Poor compliance" if the score was less than 80%. Scores were compared according to the study population characteristics. Multivariate analysis was used to identify associated factors with good practices. The threshold of statistical significance was set at p < 0.05.

### Results

We included 678 medical students. The average age was 21.76 (SD = 1.89 years) with a sex ratio of 0.40. The protection measures most respected by the participants were related to the respiratory hygiene: correct coverage of the nose and mouth with the mask (80%),

**Funding:** The author(s) received no specific funding for this work.

**Competing interests:** The authors have declared that no competing interests exist.

wearing masks regardless of the presence of symptoms (73.3%) and coverage of the mouth during coughing or sneezing (76.6%). Adherence to hand hygiene measures ranged from 51.4% to 66.3%. The least respected measures were related to social distancing: distancing of at least one meter from others (31.2%) and avoiding crowded places (42.5%). An overall score $\geq$ 80% was obtained among 61.5% of students. Referring to multivariate analysis, variables that positively affected the overall score of preventive measures related to COVID-19 were the female sex and living alone, with Beta coefficients of 3.82 and 1.37 respectively. The perceived level of stress, E-cigarette and Chicha consumption negatively affected the score with Beta coefficients of (-0.13), (-5.11) and (-2.33) respectively.

## Conclusion

The level of adherence to good practice among medical students was overall moderate. Awareness programs would be needed in this population, especially for men and those who smoke and vape.

## Introduction

Coronavirus disease 2019 (COVID-19) is a highly contagious emerging respiratory disease caused by a novel coronavirus [1]. This pandemic has led to a major health crisis which had serious global economic consequences. As of August 16, 2022, 588757628 cases have been confirmed and 6433794 COVID-19 deaths have been reported globally [2]. It is a potentially serious disease which morbidity and mortality are mainly related to the advanced age and comorbidities of patients [3].

COVID-19 most commonly gives relatively common symptoms with other types of coronavirus, such as SARS and MERS [4]. This virus is transmitted from one infected person to another through respiratory droplets and aerosols when an infected person breathes, coughs, sneezes, sings, screams or speaks [5].

Health control measures (containment and preventive practices) were imposed to reduce the risk of transmission of the virus and reduce its damage [6, 7]. Despite the vaccination that is the main solution to defeat this virus and protect the community, compliance with the preventive measures recommended by the WHO remains essential especially with the appearance of new variants of the virus. The main preventive practices recommended by the WHO were mask use, hand hygiene and social distancing [8].

In order to determine the degree of adherence to these practices, numerous studies describing Knowledge, Attitudes and Practices (KPAs) have been conducted in several countries [9–11]. Some KAP surveys have specifically targeted at-risk populations such as health professionals [12]. Others targeted the youth population such as students [13–15].

The latter, although less likely to develop serious forms of the disease, contribute significantly to the spread of the virus as a large proportion of the forms they develop remain asymptomatic or pauci-symptomatic. Health students are on the one hand a population more exposed to the virus due to close contact with the hospital environment and on the other hand a working population and can therefore be a vector and a source of viral spread.

The assessment of COVID-19 preventive practices among these students has been the subject of several studies in many countries. Medical students should be considered as highly effective partners in COVID-19 prevention efforts. They can engage meaningfully to be

educators and agents of change among their peers and in their communities. The assessment of preventive practices in this population would be of great help to know how to strengthen the adherence to good practices among them and motivate them to play a role model in the community. The aim of this study was to assess adherence to preventive behaviors related to COVID-19 and to identify its associated factors among medical students.

## Methods

### Study design

This is a cross-sectional study conducted in October 2020 at the faculty of medicine of Monastir.

### Setting

The faculty of medicine of Monastir is one of the four medical schools in Tunisia. The curriculum includes six years of undergraduate studies. The first and the $2^{nd}$ cycle of medical studies included two and four years respectively. In the 2020–2021 academic year, nearly 1,400 students were enrolled in this faculty [16].

### Study population

The survey was conducted October 21–28, 2020, during registration days for the 2020–2021 academic year which coincided with the second wave of COVID-19 in Tunisia. During the study period, the number of confirmed new cases of COVID-19 per day in the country ranged from 1300 to 1900 [17]. All undergraduate medical students of the faculty of medicine of Monastir from the first year to the fifth year were included in the study. Participants were recruited through convenience sampling.

The required sample size was calculated based on 60% adherence to Covid-19 protection measures abased on previous studies [18, 19]. The following formula was used: $n = (z_{\alpha/2})^2 \frac{p(1-p)}{i^2}$ (n: sample size, Zα/2 = 1.96, i: accuracy, 0.05, p: estimated prevalence of compliance with hygiene measures). The minimum number of participants required was 369.

### Data collection

Data were collected using an anonymous self-administered questionnaire which consisted of two sections. The first part included variables on demographic and physical health status: gender, age, household income, educational level, marital status, living alone or with others, chronic diseases, history of Covid-19 infection, smoking status (classic cigarette, E cigarette and Chicha), levels of anxiety (using the General Anxiety Disorder-7 scale (GAD-7)) and stress (using the Perceived stress scale (PSS-10)). Levels of anxiety are categorized into four categories according to the GAD-7 scale (0–4: No anxiety; 5–9: mild; 10–14: moderate and 15–21: severe). Those of stress are classified into three categories according to the PSS-10 (0–13: low; 14–26: moderate and 27–40: high).

The second part was developed based on previous literature review and contained 11 items about practices related to the protective measures adopted for COVID-19; practices related to respiratory hygiene (six items), hand hygiene (three items), and following social distancing (two items). The items were measured on a five-point Likert scale ranging from 0 (I never do this) to 4 (I do this always). The reliability of the preventive practices of the COVID-19 scale was assessed using Cronbach's Alpha Coefficient, which was 0.83, indicating the sufficient level of reliability.

## Operational definition of the variables

The overall precautionary measures score was obtained from the sum of the 11 items already described and ranged from 0 to 44; a higher score indicated better preventive behavior toward COVID-19. Practice score was then converted to a score of 0–100.

Compliance with precautionary measures related to COVID-19 was categorized, using Bloom's cut-off point, as "good" if the score was between 80 and 100%, moderate if the score was between 60 and 79%, and poor if the score was less than 60% [14, 20, 21].

## Data analysis

Data were entered and analyzed using IBM SPSS v.23.0 (IBM Corp. Released 2015. IBM SPSS Statistics for Windows, Version 23.0. Armonk, NY: IBM Corp.). Quantitative variables were presented as mean ± standard deviation or median (Interquartile ranges) as appropriate. Categorical variables were presented as frequency and percentage.

The scores were compared in terms of students' characteristics. Scores with normal distribution were compared using T test and One-way Anova otherwise they were compared using Mann-Whitney test and Kruskal-Wallis test. Multiple linear regression analysis was used to determine the relationship between the total score and students' features. Results were considered significant at a threshold of $p < 0.05$.

## Ethical considerations

Ethical approval was obtained from the Ethics Committee of the faculty of medicine of Monastir. Official support letters were obtained from the dean of the faculty of medicine of Monastir and the Directorate of School and University Medicine of Monastir to conduct the survey. Prior to data collection, the purpose of the study was explained to the participants, and they were informed that participation is voluntary and that they will also be assured of the confidentiality of the information they provide. All study methods were performed in accordance with the ethical principles of the Declaration of Helsinki.

## Results

Our study included 678 medical students. The average age was 21.76 ± 1.89 years with a sex ratio of 0.40. Most of participants were single (81.6%). Thirty six percent were at the first level of medical education. Almost 24% of the participants belonged to families whose monthly income does not exceed 2000 dinars. More than 26% were living alone and more than 14% were followed for chronic diseases. Regarding COVID-19 infection, 2.4% had previously contracted the virus and 14.2% had a family history of COVID-19 infection. The most used mask was the cloth mask (39.2%). Regarding smoking status, 7.7% were classic cigarette smokers, 8.3% were chicha consumer and 1.6% were E Cigarette users. According to the GAD-7 scale and the PSS-10, 26.7% of the students were suffering from moderate to severe anxiety levels and 78.6% had moderate to severe levels of stress. Details of the general characteristics of the study population are described in Table 1.

The protection measures most respected by the participants were related to respiratory hygiene: The correct covering of the nose and mouth with the mask (80%), the wearing of masks regardless of the presence of symptoms (73.3%) and the coverage of mouth during coughing or sneezing (76.6%). However, only 52.1% washed their cloth masks daily, 54.8% did not reuse their surgical masks two days in a row and 42.5% avoided to touch face with hands. Compliance with hand hygiene measures ranged from 51.4% to 66.3%. The least respected measures were related to social distancing: distancing of at least one meter from everyone

**Table 1. General characteristics of the study population.**

|  | n | % |
|---|---:|---:|
| **Gender** | | |
| Male | 191 | 28.2 |
| Female | 468 | 69.0 |
| NP | 19 | 2.8 |
| **Marital status** | | |
| Single | 555 | 81.9 |
| Other | 16 | 2.3 |
| NP | 107 | 15.8 |
| **Monthly household income (dinars)** | | |
| < 1000 | 45 | 6.6 |
| 1000–2000 | 162 | 23.9 |
| 2000–3000 | 176 | 25.9 |
| >3000 | 178 | 26.3 |
| NP | 117 | 17.3 |
| **Living** | | |
| Alone | 178 | 26.2 |
| With colleagues or family | 442 | 65.1 |
| NP | 58 | 8.7 |
| **Repeating a year** | | |
| No | 476 | 70.2 |
| Yes | 32 | 4.7 |
| NP | 170 | 25.1 |
| **Chronic condition** | | |
| No | 543 | 80.1 |
| Yes | 97 | 14.3 |
| NP | 38 | 5.6 |
| **Personal history of Covid-19 infection** | | |
| No | 647 | 95.4 |
| Yes | 16 | 2.4 |
| NP | 15 | 2.2 |
| **Family history of Covid-19 infection** | | |
| No | 556 | 82 |
| Yes | 96 | 14.2 |
| NP | 26 | 3.8 |
| **Most used mask** | | |
| Surgical mask | 147 | 21.7 |
| FFP2 | 9 | 1.3 |
| Cloth mask | 266 | 39.2 |
| Many types | 229 | 33.8 |
| NP | 27 | 4 |
| **Conventional cigarettes smoking** | | |
| No | 594 | 87.6 |
| Yes | 52 | 7.7 |
| NP | 32 | 4.6 |
| **Chicha smoking** | | |
| No | 622 | 91.7 |
| Yes | 56 | 8.3 |

(*Continued*)

**Table 1.** (Continued)

|  | n | % |
|---|---:|---:|
| **E-cigarette use** |  |  |
| No | 667 | 98.4 |
| Yes | 11 | 1.6 |
| **Stress** |  |  |
| Low | 121 | 17.9 |
| Moderate | 473 | 69.8 |
| High | 60 | 8.9 |
| NP | 24 | 3.4 |
| **Anxiety** |  |  |
| No | 260 | 38.3 |
| Mild | 181 | 26.7 |
| Moderate | 123 | 18.1 |
| Severe | 58 | 8.6 |
| NP | 56 | 8.3 |

NP: Not precised.

(31.2%) and avoiding crowded areas (42.5%). Fig 1 shows the percentage of participants responding "always" for each practice assessed.

By summing up the scores of all items, the total score was 35.33 (SD:7.62, range from 4 to 44) (Table 2).

After conversion to a score of 0–100, 61.5% of participants had a score $\geq$ 80% and were considered compliant with preventive behaviors towards COVID-19 (Fig 2).

Practice scores varied significantly across gender (p $< 10^{-3}$), marital status (p = 0.034), the level of perceived stress (p = 0.04) and the smoking status (classic cigarette (p = 0.04), Chicha (p $< 10^{-3}$), and E cigarette smoking (p = 0.006)) (Table 3).

The distribution of scores by smoking status is described in Fig 3. The lowest scores were recorded among students who use chicha (33 [27–39]), E cigarette (35 [31– 40]) and especially those who use several forms of tobacco at the same time (31 [23–38.5]).

According to the multiple linear regression analysis, the variables that positively affected the overall score of precautionary measures were female gender and living alone which increased the score by 3.82 and 1.37 points respectively. E cigarette consumption and Chicha smoking reduced the score by 5.11 and 2.33 points respectively. Perceived stress also negatively affected this score (Table 4).

## Discussion

### Key results

According to our findings, compliance with good practices rates among medical students were generally moderate. Three medical students out of five (61.5%) were considered as compliant with preventive behaviors towards COVID-19 (score $\geq$ 80%). Compliance with respiratory hygiene practices was better than hand hygiene practices and social distancing. Five factors influencing the overall score of preventive practices among medical students were identified. Female sex and living alone positively affected this score with Beta coefficients of (3.82) and (1.37) respectively. The use of Chicha, the use of E-cigarettes and the perceived level of stress negatively affected this score with Beta coefficients of (-5.11), (-2.33) and (-0.13) respectively.

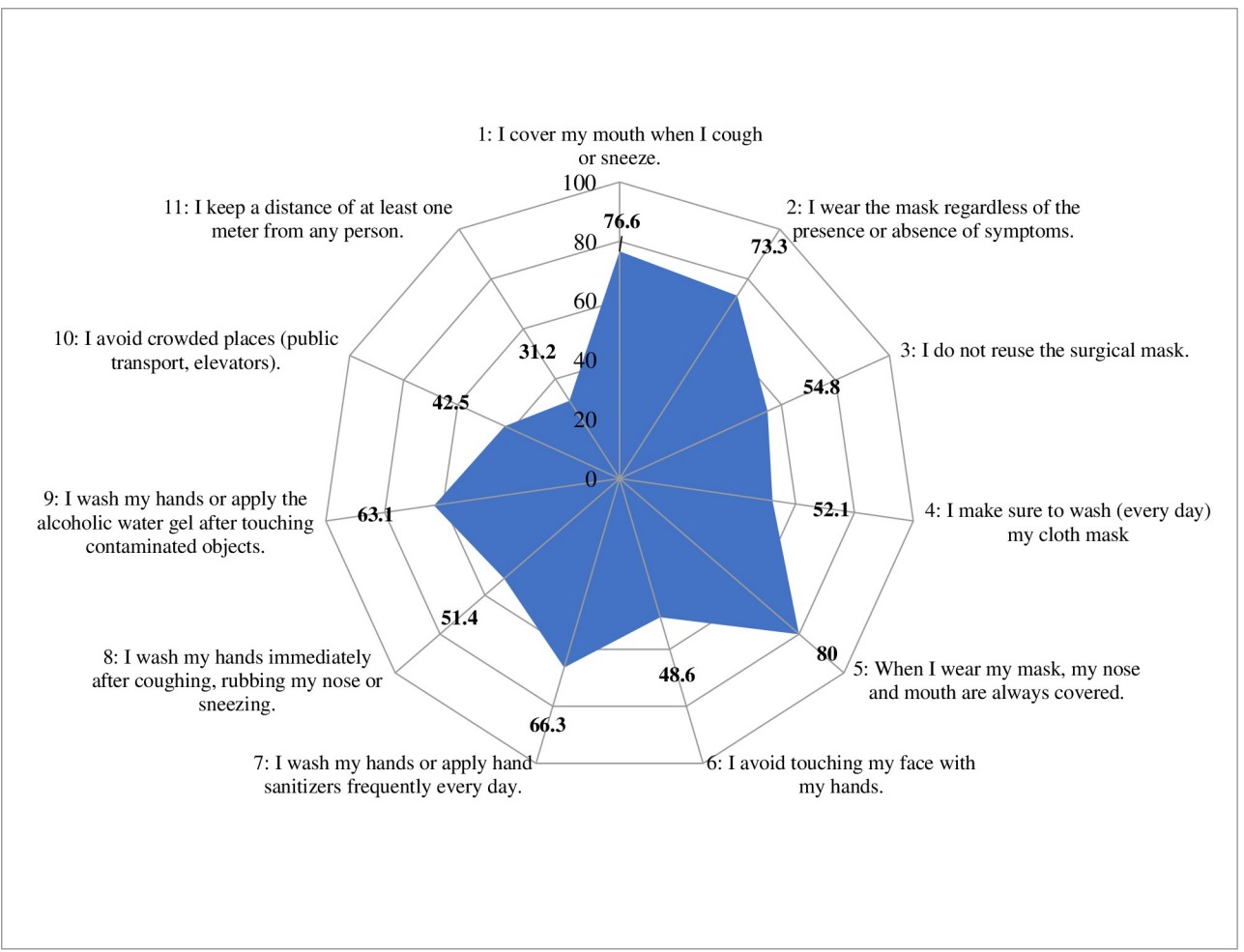

**Fig 1. Frequency of "Always" response to the precautionary measures' items.**

**Table 2. Average scores of preventive practices among medical students at the medical school of Monastir.**

| | N | Mean | SD |
|---|---|---|---|
| 1: I cover my mouth when I cough or sneeze. | 674 | 3.55 | 0.94 |
| 2: I wear the mask regardless of the presence or absence of symptoms. | 674 | 3.55 | 0.90 |
| 3: I do not reuse the surgical mask. | 670 | 2.92 | 1.41 |
| 4: I make sure to wash (every day) my cloth mask | 658 | 3.10 | 1.15 |
| 5: When I wear my mask, my nose and mouth are always covered. | 670 | 3.60 | 0.88 |
| 6: I avoid touching my face with my hands. | 673 | 3.17 | 0.99 |
| 7: I wash my hands or apply hand sanitizers frequently every day. | 668 | 3.46 | 0.88 |
| 8: I wash my hands immediately after coughing, rubbing my nose or sneezing. | 670 | 3.16 | 1.06 |
| 9: I wash my hands or apply the alcoholic water gel after touching contaminated objects. | 669 | 3.40 | 0.94 |
| 10: I avoid crowded places (public transport, elevators). | 671 | 3.06 | 1.03 |
| 11: I keep a distance of at least one meter from any person. | 667 | 2.75 | 1.07 |
| **Questions Sum Scores/44** | 678 | 35.33 | 7.62 |
| **Mean total score /4** | 678 | 3.21 | 0.69 |

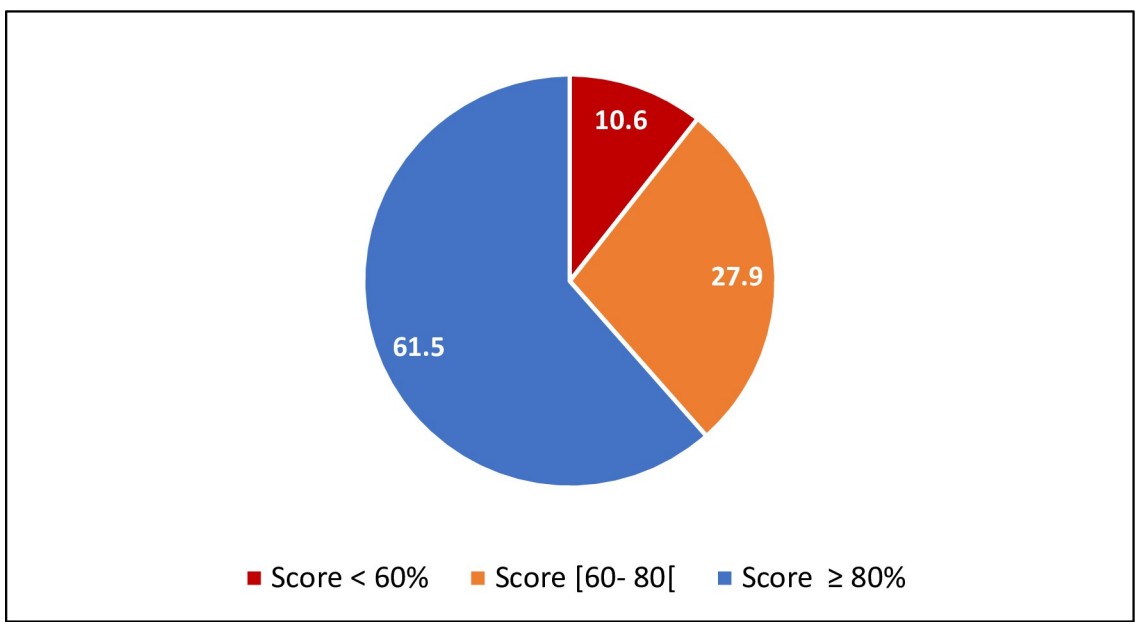

**Fig 2. Profile of medical students' practices score.**

### Interpretation

The results of our study showed that 61.5% of students at the faculty of medicine of Monastir complied with COVID-19 protection measures. A similar study conducted by Ketata et al. at the faculty of medicine of Sfax in Tunisia showed that this prevalence was 61.3% [19]. These practices were also evaluated in Algeria among students from several fields, revealing that only 58.6% complied with good practices [18]. This prevalence was 43.5% in Indonesia according to a multicenter study conducted online and targeted medical students [14] while according to the results of a survey conducted among students and the academic and administrative staff of a university in Malaysia, more than 80% (85.5–100%) of respondents respected standard COVID-19 preventative measures [22]. In Vietnam, a study among health science students found that the adherence rate to good practice was 92.8% [23].

This variation may be related to the intrinsic characteristics of the study population but may also be due to the difference in the instruments used to evaluate the students' practice as well as the methods and thresholds for categorizing good and bad practices; some studies have set thresholds of 80%, in others the threshold was 70%, while other studies have used the average practice score as a limit value to dichotomize bad and good practices [24, 25].

According to other authors, the results were represented as average scores or total scores of practices without categorization into good and bad practices. The quantitative analysis of our results found an average practice score of 3.2/4 (± 0.69). This score was 3.7/4 (± 0.6) among health science students from 25 Egyptian universities according to Salem et al. [15]. The study of Zhao et al. [26] assessing these practices among students on a 45 scale and conducted in East Asia showed total scores of 33.9 (± 5.28), 39.30 (± 4.46) and 34.39 (± 5.53) respectively in Korea, China and Japan.

### Respiratory hygiene measures

According to our findings, the most respected protective measures were related to respiratory hygiene: proper coverage of the nose and mouth with the mask (80%), wearing masks

**Table 3. The relationship between the study population characteristics and precautionary measures score.**

|  | N | Total Score | p |
|---|---|---|---|
| **Gender** |  |  |  |
| Male | 191 | 32.27 ± 8.90 | $<10^{-3}$ |
| Female | 468 | 36.89 ± 6.25 |  |
| **Marital status** |  |  |  |
| Single | 555 | 38 [33–40] | **0.034*** |
| Other | 16 | 32 [16.50 - 39.75] |  |
| **Monthly household income (dinars)** |  |  |  |
| < 2000 | 207 | 35.56 ± 7.09 | 0.79 |
| ≥2000 | 354 | 35.38 ± 7.86 |  |
| **Living** |  |  |  |
| Alone | 178 | 36.43 ± 7.26 | 0.11 |
| With colleagues or family | 442 | 35.43 ± 7.15 |  |
| **Repeating a year** |  |  |  |
| Yes | 32 | 33.46 ± 9.04 | 0.13 |
| No | 476 | 35.99 ± 6.76 |  |
| **Conventional cigarettes smoking** |  |  |  |
| Yes | 52 | 33.09 ± 8.64 | **0.04** |
| No | 594 | 35.61 ± 7.38 |  |
| **Chicha smoking** |  |  |  |
| Yes | 56 | 30.28 ± 9.94 | $<10^{-3}$ |
| No | 622 | 35.78 ± 7.22 |  |
| **E-cigarette use** |  |  |  |
| Yes | 11 | 31 [13–35] | **0.006*** |
| No | 667 | 37 [32–40] |  |
| **Anxiety** |  |  |  |
| None or Mild | 441 | 35.98 ± 7.13 | 0.25 |
| Moderate or severe | 181 | 35.27 ± 6.85 |  |
| **Stress** |  |  |  |
| Low | 121 | 36.72 ± 7.27 | **0.04** |
| Moderate or high | 473 | 35.23 ± 7.45 |  |
| **Chronic condition** |  |  |  |
| Yes | 97 | 35.01 ± 8.11 | 0.30 |
| No | 543 | 35.83 ± 7.03 |  |
| **Personal history of Covid-19 infection** |  |  |  |
| Yes | 16 | 36.5 [31–40] | 0.52* |
| No | 647 | 38 [32–40] |  |
| **Family history of Covid-19 infection** |  |  |  |
| Yes | 96 | 36.15 ± 5.99 | 0.54 |
| No | 556 | 35.69 ± 7.12 |  |

Scores with normal distribution were described as Mean ± SD otherwise they were described as Median [IQR].
*Mann-Whitney Test.

regardless of the presence of symptoms (73.3%) and coverage of the mouth during coughing or sneezing (76.6%). This last practice has always been respected only by 53.4% of students according to the Algerian study of Oualid et al. [18] while in India the studies of Prasard et al. [27] and Maheshwari et al. [28] found that this practice was respected by the vast majority of students (97.4% and 96.9% respectively).

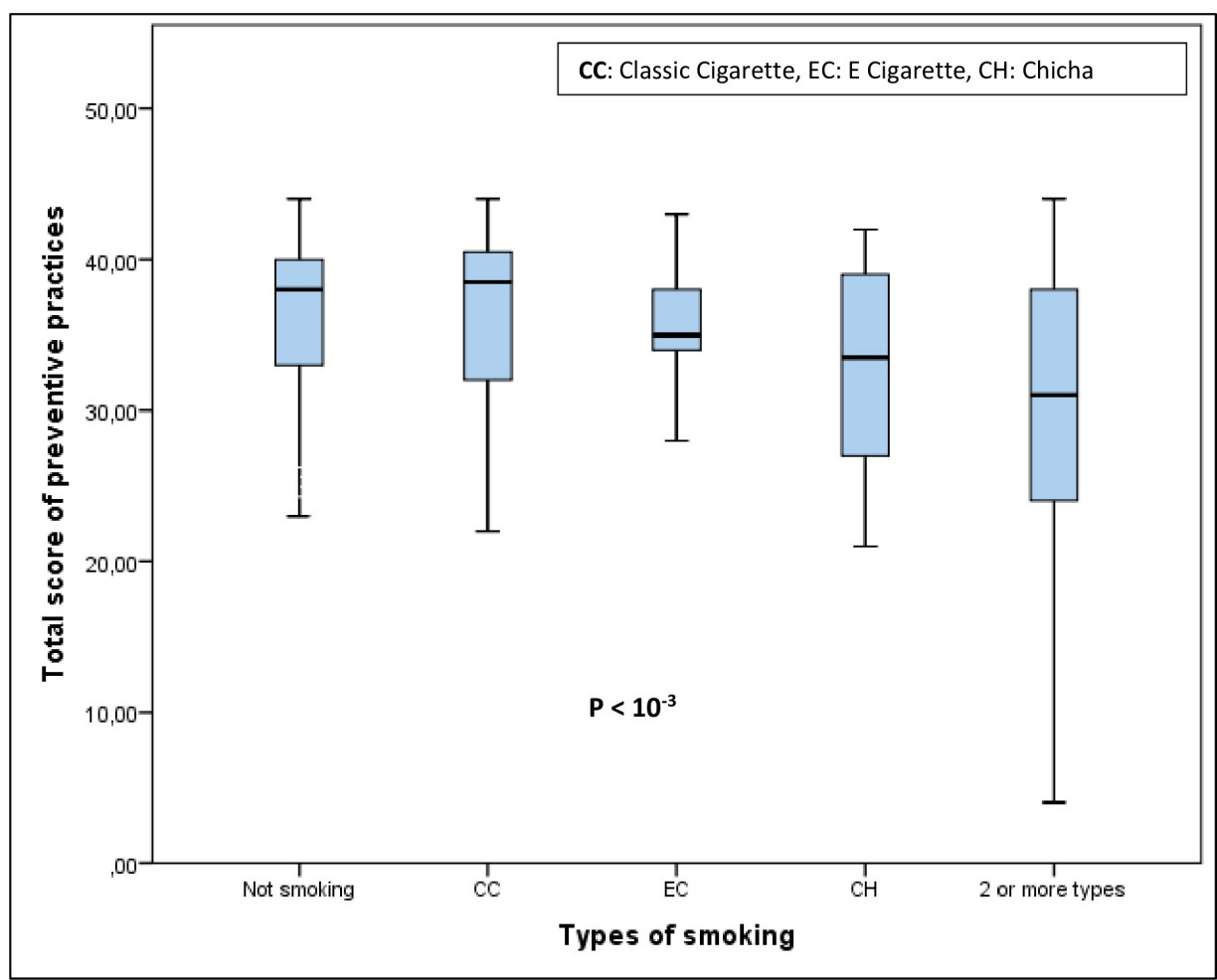

**Fig 3. Comparison of the precautionary measures score according to the smoking status.**

**Table 4. Associated factors with adherence to preventive behaviors related to COVID-19 among medical students (Multiple regression analysis).**

| | β | SE | p | IC 95% | |
|---|---|---|---|---|---|
| Constant | 31.65 | 1.39 | $< 10^{-3}$ | 28.91 | 34.40 |
| Female gender [a] | 3.82 | 0.62 | $< 10^{-3}$ | 2.59 | 5.04 |
| Chicha smoking [b] | - 2.33 | 1.06 | **0.03** | - 4.42 | - 0.24 |
| Electronic cigarette use [c] | -5.11 | 2.16 | **0.01** | - 9.36 | - 0.86 |
| Living alone [d] | 1.37 | 0.60 | **0.02** | 0.19 | 2.55 |
| Percieved stress score [e] | -0.13 | 0.04 | **0.004** | -0.22 | -0.04 |

a. Reference category: Male Gender.

b. Reference category: No Chicha smoking.

c. Reference category: No E-Cigarette use.

d. Reference category: Living with colleagues or family.

e. Quantitative variable.

Masking was evaluated among students through different questions depending on the studies: It was only respected among less than 20% of university students in Pakistan [29]. In the Algerian study of Oualid et al., the wearing of masks was always respected in 41.3% of students [18]. In Jordan, a KAP survey targeting students from different medical and non-medical fields showed that 64.7% of them wore masks [30]. In Egypt, the study of Salem MR et al. conducted among medical students found that 93.4% of respondents wore the mask outside the house [15]. This percentage was 100% among students and academic staff according to Lee KM et al. in Malaysia [22]. The correct coverage of the mouth and nose by the mask was respected in 90% of cases according to a study in Vietnam targeting students in health sciences [23].

## Hand hygiene measures

Our results showed that compliance with hand hygiene measures ranged from 51.4% to 66.3% among students in the faculty of medicine of Monastir. It was only of 44.2% among students in Algeria [18]. This rate was slightly higher (64.2%), according to the study of Angelo AT et al. conduct in Ethiopia [24]. The study carried out in Vietnam by Le An et al. showed that almost 95% of students washed their hands frequently every day with an average duration of at least 20 seconds [23]. This rate was 85.5% in the study conducted in Malaysia [22]. In India, the washing of hands with soap or the use of hydroalcoholic solutions was respected among students with compliance rates exceeding 90% [27, 28]. In Egypt and Jordan, compliance with these practices was considered satisfactory and was 92.2% and 98.9% respectively in these two countries [15, 30].

## Social distancing

Our study showed that the least respected measures were related to social distancing: social distancing of one metre at least (31.2%) and avoiding crowded areas (42.5%). In Algeria, these rates were respectively 21.4% and 24.4% [18]. In Ethiopia, the majority (60.6%) did not respect social distancing, and 61.4% attended places where there was a gathering of people [24]. In India, distancing was respected by 98.3% of medical students [28]. In Vietnam, social distancing as directed by the Ministry of Health was respected by 96.2% of students surveyed. This same study also showed that the eviction of unnecessary travel and crowded places were well respected by these students (97.3%) [23]. Levels of compliance with this practice exceeding 90% among health students were also reported in Egypt (91.2%) [15] and Malaysia (95.9%) [22]. Discrepancies reported between studies may be explained by differences in the characteristics of the populations. Indeed, according to the multicentric study by Zhao et al. [26] conducted online and targeted three populations in East Asia (China, Korea and Japan), the most respected practice by the Chinese population was social distancing whereas in the Japanese and Korean populations, compliance with hand washing practice was better. These differences could also be explained by the different methodology of the surveys, especially the way in which the questionnaire was administered (online, face-to-face, self-administered, etc.). In fact, the data collected via a face-to-face questionnaire may be less objective than those obtained through a self-administered questionnaire and may lead to overestimated levels of compliance with good practices. Furthermore, online questionnaires cannot reach the entire target population. Only students with internet access can participate in the survey. The epidemiological situation at the time of the survey could also explain this variability in results. Indeed, the practices at the very beginning of the epidemic are not the same as those during the peaks of the epidemic.

## Compliance with good practices

According to other studies conducted among health professionals, compliance with good practices was better among them. Indeed, the study by Boujamline et al. [31] conducted at a university Hospital in Tunisia found that 94% of the staff wore personal protective equipment. A study of health professionals in China found that the proportion of people wearing masks in public places was 93.01%; social distancing of at least one metre was respected by 72.03%; hand washing three times a day was respected by 96.5% and 89.51% of staff wore disposable medical masks at work [12]. However, some studies targeting the general population have shown lower levels of adherence to these practices. In Saudi Arabia, Bin Abdulrahman AK et al. [32] found that only 23% of participants in a KAP survey targeting the Saudi population washed their hands between 20 and 30 s and 59.6% washed their hands after shaking hands with others. Also in Syria, masks were only respected by 27.9% of participants according to an online survey targeting the general population in Syria [33].

## Factors associated with good practices

Several factors were explored in this study to identify determinants of good practices among medical students. The female gender was a predictor of good practice in our study. This finding is consistent with other studies that evaluated COVID-19 prevention practices among students [14, 15, 21, 26]. This could be explained by a better motivation for learning among female students as explained by Ghazvini et al. [34], leading to better compliance with good practice. Several other studies targeting health professionals [35] and the general population have also confirmed this link. This relationship has been proven even in other epidemic contexts. Indeed, a meta-analysis published in 2016 in the context of epidemics such as influenza, showed that women practice non-pharmacological protective behaviour more than men [36]. The same finding is also consistent with other studies examining the influence of gender on practices related to other health conditions [37, 38]. However, some studies have not found this association [39, 40]. Regarding the relationship with tobacco consumption, we found that adherence to preventive practices varied according to the form of tobacco use: the use of chicha negatively affected the score of good practices. Indeed, the consumer of chicha finds himself obliged to spend a long time without a mask to smoke the chicha. On the other hand, in most cases, people dependent on chichas do not have a personal chicha and find themselves obliged to smoke it in crowded places (tea rooms, cafe, etc.) where social distancing is not respected. Furthermore, our results have shown that vaping has also been predictive of poor compliance with good practices. However, the multivariate analysis of our study did not reveal any association between the consumption of conventional cigarettes and the level of compliance with preventive measures. This is consistent with the results of Angelo AT et al. [24] where no association between smoking and the level of adherence to good practice was noted. However, according to Hosen et al. [13], smoking and alcohol use were identified as predictors of poor practice. These findings could be attributed to a lower level of health concern among people who use tobacco and alcohol.

Regarding housing, according to our findings, students living alone were more likely to apply COVID-19 prevention rules. This finding was not demonstrated in either Le An et al. [23] or Shewale et al. [39] studies where compliance with good practice has not been influenced by the number of people living in the same house. In addition, the study of Lee et al. [22] showed that living with children, was positively associated with compliance with hygiene measures, however living with elderly had no impact on the degree of compliance with these rules. Other studies have discussed habitat location (rural or urban); the Angelo et al. study found that living in a rural setting was positively associated with the degree of adherence to

COVID-19 precautions. In contrast, the Hatabu et al. [41] study in Japan showed that living in an urban environment was associated with better adherence to good practice. Hosen et al. in Bangladesch [13] found that village life was associated with poor adherence to good practice. However, Oualid et al. in Algeria [18] did not find any relationship between these preventive practices and the habitat (rural or urban).

The relationship with stress and anxiety levels were also explored in the literature. Our study found that being stressed improves students' adherence to COVID-19 preventative measures, but anxiety has no impact. The study by Shewale et al. [39] found no relationship with stress or anxiety. Similarly, for the study by Bin Abdulrahman et al. [32] conducted in Saudi Arabia, no correlation between participants' level of stress and their compliance with hygiene rules was found.

Association with other factors such as income, marital status, Covid-19 and chronic disease history was explored in this study but we concluded that there was no association with the degree of compliance with good practice. However, the results in the literature were divergent.

## Limitations

Our study has some limitations that should be mentioned: First, the type of non-random sampling could alter the representativeness of the study population. Second, the assessment of adherence to preventive measures was not based on objective observations, but on a self- administered questionnaire which could result in information bias. Third, the cross-sectional design of the study did not allow the evaluation of practices according to the evolution of the epidemic.

## Conclusion

In conclusion, our findings have shown that adherence levels to COVID-19 prevention practices among medical students were generally moderate. The most respected protective measures by participants were related to respiratory hygiene. The least respected measures were related to social distancing. Male gender, chicha smoking, vaping, and using different forms of tobacco at the same time were predictors of poor adherence to these precautionary measures. The results of this study would be very useful for the future since we have been seeing more and more epidemics of communicable diseases (coronavirus, influenzae, monkeypox. . .). Thus, to improve the level of adherence to precautionary practices related to these communicable diseases among students, awareness programs would be needed, with a particular focus on men and those who smoke and vape. These programs should also aware medical students about their role model for the society and motivate them to be partners in epidemic prevention efforts.

## Supporting information

**S1 File. French version of the Perceived Stress Scale (PSS-10).**
(PDF)

**S2 File. French version of the General Anxiety Disorder scale (GAD-7).**
(PDF)

**S3 File. Associated factors with adherence to preventive behaviors related to COVID-19 among medical students in the university of Monastir, Tunisia.**
(XLSX)

## Acknowledgments

We wish to thank the dean of the faculty of medicine of Monastir and the Director of the School and University Medicine of Monastir for allowing us to conduct this study. We would

like also to thank the administrative staff of the faculty of medicine of Monastir for their help during the survey. And we take this opportunity to extend our gratitude to all the team of the Department of Epidemiology and Preventive Medicine at Monastir University for their implication in data collection.

## Declarations

### Ethics approval and consent to participate

The study was conducted under Good Clinical Practice conditions and according to ethical standards collections. All methods were performed in accordance with the relevant guidelines and regulations. Written Informed consent to participate was obtained from all study participants. The ethics committee of Faculty of medicine of Monastir approved the protocol which was performed according to the principles of the Helsinki Declaration of 1983.

## Author Contributions

**Conceptualization:** Imen Zemni, Asma Belguith Sriha.

**Formal analysis:** Imen Zemni, Kaouthar Zemni, Amal Gara, Amani Maatouk, Manel Ben Fredj, Hela Abroug, Meriem Kacem, Cyrine Benasrallah, Wafa Dhouib.

**Methodology:** Imen Zemni.

**Supervision:** Asma Belguith Sriha.

**Validation:** Asma Belguith Sriha.

**Writing – original draft:** Kaouthar Zemni.

**Writing – review & editing:** Imen Zemni, Ines Bouanene, Asma Belguith Sriha.

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
