## [Decision Letter · Decision Letter 0]

8 Nov 2022

PONE-D-22-24560Associated factors with adherence to preventive behaviors related to COVID-19 among medical students in the university of Monastir, TunisiaPLOS ONE

Dear Imen Zemni

Thank you for submitting your manuscript to PLOS ONE. After careful consideration, we feel that it has merit but does not fully meet PLOS ONE’s publication criteria as it currently stands. Therefore, we invite you to submit a revised version of the manuscript that addresses the points raised during the review process.

We look forward to receiving your revised manuscript.

Kind regards,

Muhammad Junaid Farrukh

Academic Editor

PLOS ONE

Journal Requirements:

Reviewers' comments:

Reviewer's Responses to Questions

**Comments to the Author**

1. Is the manuscript technically sound, and do the data support the conclusions?

Reviewer #1: Yes

Reviewer #2: Yes

2. Has the statistical analysis been performed appropriately and rigorously? 

Reviewer #1: Yes

Reviewer #2: Yes

3. Have the authors made all data underlying the findings in their manuscript fully available?

Reviewer #1: Yes

Reviewer #2: No

4. Is the manuscript presented in an intelligible fashion and written in standard English?

Reviewer #1: Yes

Reviewer #2: No

5. Review Comments to the Author

Reviewer #1: Dear authors, very good work. Only some simple points:

Methods:

1. The phrase” Were included in the study, all undergraduate medical students of the

faculty of medicine of Monastir from the first year to the fifth year.” on page 13, line 4, needs to be rearranged linguistically. In my opinion, it can be changed into “All undergraduate medical students of the faculty of medicine of Monastir from the first year to the fifth year were included in the study.”

2. I suggest to mention the scores of the GAD 7 scale and PSS-10 in methods.

3. Are the scores of the adherence only good (80-100%) or bad (less than 80%), not moderate?

Results:

In table 1:

a. The sum of percentages of marital status is 99.77, not 100%, you must modify?

b. The sum of percentages of stress is 99.9, not 100%, you must modify?

Reviewer #2: I appreciate your efforts in this study. Generally speaking the work is technically acceptable, and the defects could be amended in the review process. However, considering the rapidly evolving the public knowledge and practice towards COVID-19, I'm afraid publishing a survey which is conducted 2 years ago would be futile. It's very likely that the same respondents' current practice and knowledge is very different from 2 years ago. This study would neither reflect the current situation nor assist policy makers in making sound, updated decisions. I wish the authors could publish the work shortly after the study was performed. A delay in publishing surveys on many chronic conditions can be ignored, as the public perceptions and practice may vary slowly over time. But KAP studies about an infectious pandemic, especially when the causative agent, and subsequently the disease manifestations and our knowledge about that, are highly variating, should be published as soon as possible. Otherwise, the benefits of disseminating the findings of the study would not be achieved.

6. PLOS authors have the option to publish the peer review history of their article (what does this mean?). If published, this will include your full peer review and any attached files.

Reviewer #1: **Yes: **Rami Abduljabbar, The University of Jordan, Amman, Jordan.

Reviewer #2: No

---

## [Author Response · Author response to Decision Letter 0]

23 Dec 2022

Dear Editor,

Thank you for your attention to our work. The comments and suggestions we have received are valuable and very helpful to improve our manuscript. We have made revisions based on your latest comments and suggestions, as described in the authors' response.

We respond in detail to the reviewers’ comments. They raised important issues and we agree with almost all their comments. We have revised our manuscript according to their indications. We hope that they will find our responses to their comments satisfactory, and we are willing to finish the revised version of the manuscript including any further suggestion that the reviewers may have. The revision has been developed in consultation with all coauthors, and each author has given approval to the final form of this revision. All changes made in the revised version will be visible in red.

Please find as attached files:

The “Response to Reviewers”,

The “Revised Manuscript with Track Change” and

the “Manuscript” (the unmarked version of the revised manuscript).

We sincerely hope that the revision of the manuscript will be satisfactory, and the enclosed version will be acceptable for publication.

Once again thank you for your cooperation.

Sincerely Yours,

Dr Imen ZEMNI

 

Journal Requirements:

Response: Thank you, we have done the required modifications. 

2. In your Data Availability statement, you have not specified where the minimal data set 

underlying the results described in your manuscript can be found. PLOS defines a study's minimal data set as the underlying data used to reach the conclusions drawn in the manuscript and any additional data required to replicate the reported study findings in their entirety. All PLOS journals require that the minimal data set be made fully available. 

Response: Thank you, we have uploaded data as a supporting information file (S3. File).

3. Please review your reference list to ensure that it is complete and correct. If you have cited 

papers that have been retracted, please include the rationale for doing so in the manuscript text or remove these references and replace them with relevant current references. Any changes to the reference list should be mentioned in the rebuttal letter that accompanies your revised manuscript. If you need to cite a retracted article, indicate the article’s retracted status in the References list and also include a citation and full reference for the retraction notice.

Response: Thank you, we have made the necessary revisions.

Review Comments:

Reviewer #1: 

Dear authors, very good work. Only some simple points:

Methods:

1. The phrase” Were included in the study, all undergraduate medical students of the faculty of medicine of Monastir from the first year to the fifth year.” on page 13, line 4, needs to be rearranged linguistically. In my opinion, it can be changed into “All undergraduate medical students of the faculty of medicine of Monastir from the first year to the fifth year were included in the study.” 

Response: Thank you, we have done the required modifications. (Lines 115-116)

2. I suggest to mention the scores of the GAD 7 scale and PSS-10 in methods. 

Response: Thank you, we have added supporting files detailing the GAD 7 scale and the PSS-10 scale (S1. File and S2. File) and we have better detailed the interpretation of the scores. (Lines 128 -131)

4. Are the scores of the adherence only good (80-100%) or bad (less than 80%), not moderate? Response: Thank you, we have categorized compliance with precautionary measures related to COVID-19 using Bloom’s cut-off point, as “good” if the score was between 80 and 100%, moderate if the score was between 60 and 79%, and poor if the score was less than 60%. We have added these details in the methods section (Lines 143-145) and we have represented the results of these three classes in Figure 2.

Results:

In table 1:

a. The sum of percentages of marital status is 99.77, not 100%, you must modify?

b. The sum of percentages of stress is 99.9, not 100%, you must modify?

Response: Thank you, we have made the necessary corrections (Table 1).

Reviewer #2: 

I appreciate your efforts in this study. Generally, speaking, the work is technically acceptable, and the defects could be amended in the review process. However, considering the rapidly evolving the public knowledge and practice towards COVID-19, I'm afraid publishing a survey which is conducted 2 years ago would be futile. It's very likely that the same respondents' current practice and knowledge is very different from 2 years ago. This study would neither reflect the current situation nor assist policy makers in making sound, updated decisions. I wish the authors could publish the work shortly after the study was performed. A delay in publishing surveys on many chronic conditions can be ignored, as the public perceptions and practice may vary slowly over time. But KAP studies about an infectious pandemic, especially when the causative agent, and subsequently the disease manifestations and our knowledge about that, are highly variating, should be published as soon as possible. Otherwise, the benefits of disseminating the findings of the study would not be achieved.

Response: Thank you, we agree with you.

We have tried to explain the impact and usefulness of this study for the future (lines 388 - 394):

We think that the results of this study would be very useful for the future since we have been seeing today more and more epidemics of communicable diseases (coronavirus, influenzae, monkeypox...). 

Our study has identified categories with poor adherence to precautionary measures among medical students. Awareness programs should target especially these students in future epidemic contexts.

---

## [Editor Report · Decision Letter 1]

12 Jan 2023

Associated factors with adherence to preventive behaviors related to COVID-19 among medical students in the university of Monastir, Tunisia

PONE-D-22-24560R1

Dear Imen Zemni

We’re pleased to inform you that your manuscript has been judged scientifically suitable for publication and will be formally accepted for publication once it meets all outstanding technical requirements.

Kind regards,

Muhammad Junaid Farrukh

Academic Editor

PLOS ONE

Additional Editor Comments (optional):

Reviewers' comments:

<quillbot-extension-portal></quillbot-extension-portal>

---

## [Editor Report · Acceptance letter]

3 Mar 2023

PONE-D-22-24560R1 

Associated factors with adherence to preventive behaviors related to COVID-19 among medical students in the university of Monastir, Tunisia 

Dear Dr. Zemni:

I'm pleased to inform you that your manuscript has been deemed suitable for publication in PLOS ONE. Congratulations! Your manuscript is now with our production department. 

Kind regards, 

on behalf of

Dr. Muhammad Junaid Farrukh 

Academic Editor

PLOS ONE